# Anxiolytic Effects of *Acanthopanax senticosus* HARMS Occur via Regulation of Autonomic Function and Activate Hippocampal BDNF–TrkB Signaling

**DOI:** 10.3390/molecules24010132

**Published:** 2018-12-31

**Authors:** Shouhei Miyazaki, Hirotaka Oikawa, Hideo Takekoshi, Masako Hoshizaki, Masato Ogata, Takahiko Fujikawa

**Affiliations:** 1Laboratory of Molecular Prophylaxis and Pharmacology, Graduate School of Pharmaceutical Sciences, Suzuka University of Medical Science, 3500-3 Minamitamagaki-cho, Mie 513-8670, Japan; dp17002@st.suzuka-u.ac.jp; 2Faculty of Pharmaceutical Sciences, Suzuka University of Medical Science, 3500-3 Minamitamagaki-cho, Mie 513-8670, Japan; oikawah@suzuka-u.ac.jp; 3Sun Chlorella Corp., Production & Development Department, 369 Osaka-cho, Karasuma-dori Gojo-sagaru, Shimogyo-ku, Kyoto 600-8177, Japan; htakekoshi@sunchlorella.co.jp (H.T.); mhoshizaki@sunchlorella.co.jp (M.H.); 4Department of Biochemistry and Proteomics, Mie University Graduate School of Medicine, 2-174 Edobashi, Tsu, Mie 514-8507, Japan; ogata@doc.medic.mie-u.ac.jp

**Keywords:** *Acanthopanax senticosus* HARMS, anti-anxiety, BDNF, parasympathetic, sympathetic

## Abstract

Mental stress, such as anxiety and conflict, causes physiological changes, such as changes in autonomic nervous activity and gastric ulcers. In addition, stress induces glucocorticoids and changes the hippocampal brain-derived neurotrophic factor (BDNF) expression levels. We previously reported that *Acanthopanax senticosus* HARM (ASH) prevents stress-induced gastric ulcers. Thus, we investigated the potential anxiolytic effect and influence of ASH on the hippocampus BDNF-related protein in male Sprague-Dawley rats fed 1% and 5% ASH extract-containing food for one week using novelty suppressed feeding (NSF) and improved elevated beam walking (IEBW) tests. ASH treatment significantly decreased latency to eat in the NSF test and increased the time spent on the open arm in the IEBW test. ASH5% treatment showed a significant decrease in LFnu, indicative of sympathetic nervous activity, and a significant increase in HFnu, indicative of parasympathetic nervous activity, in the NSF test. In addition, ASH1% and ASH5% treatments significantly decreased LFnu and significantly increased HFnu in the IEBW test. ASH5% treatment significantly increased hippocampal BDNF protein expression in both Western blotting and immunohistochemistry experiments. Our findings suggest that anxiolytic effects of ASH occur via the regulation of autonomic function and increased hippocampal BDNF signaling.

## 1. Introduction

*Acanthopanax senticosus* HARMS (ASH) is a deciduous shrub and a member of the *Araliaceae* family that grows abundantly in various regions of Russia, China, Korea, Southeast Asia, and north Japan. In China, ASH root bark is traditionally used to treat high blood pressure, mental disorders, and rheumatoid arthralgia [1,2,3,4]. In Western countries, ASH is widely used as an alternative medicine. A recent study was conducted on the importance of ASH root and stem barks, as well as ASH fruit and leaf. The ASH fruit improves insulin resistance and hepatic lipid accumulation [5]. ASH leaves have antihyperlipidemic effects [6] in high-fat diet-fed mice and gastroprotective effects in acute gastric mucosal lesion in rats [7]. ASH is also an adaptogen and increases stress resistance [8,9,10,11]. ASH has a protective effect against gastric ulcers [12] and prevents depressive symptoms induced by tail suspension and the forced swimming test [13,14,15].

A recent study showed that bioactive components, including syringing, chlorogenic acid, eleutheoside E, and isofraxidin, in ASH, and ASH extracts are characterized for their pharmacological effects [16]. Syringin (also referred to eleutheroside B) was considered to have anti-inflammatory effect and improve insulin resistance [17,18]. Eleutheroside E was reported to have anti-inflammatory effects [19,20], protective effects for heart ischemia [21], and effects for improving insulin resistance [22]. Isofraxidin was reported to have anticancer effects [23] and protective effects against lipid metabolism disorder induced by a high-fat diet [24]. Chlorogenic acid is well-known as an antioxidant [25] and was indicated to have various effects on health [26].

Changes in the external environment cause stress, which affects the hypothalamus and induces gastric ulcers and depression via the sympathetic-adrenergic and pituitary-adrenal cortex systems [27,28]. Anxiety and depressive-related behaviors are observed in rats after myocardial infarction and are associated with a decline in the autonomous control of the heart rate [29]. Sympathetic nervous system (SNS) activation increases heart rate, cardiac stroke volume, and vasoconstriction, which are consistent with somatic tension. In contrast, parasympathetic nervous system (PNS) activation induces the opposite effect and alleviates somatic tension. Accordingly, anxiety induces SNS activation, while decreasing that of the PNS [30]. This phenomenon was demonstrated in anxiety disorder patients and rodents [31,32,33]. These findings suggest that psychogenic stress resistance by ASH is caused by affecting the central nervous system (CNS). However, these potential effects remain unclear. Therefore, analyzing the effect of ASH on the autonomic nervous system (ANS) is also important.

In humans and rodents, decreasing hippocampal brain-derived neurotrophic factor (BDNF) concentrations may be involved in the onset of depression and anxiety. The hippocampus is considered to be involved in learning, memory, and anxiety. BDNF plays an important role in neurogenesis and synaptic plasticity, as well as cognition and mood [34,35,36,37]. Accordingly, stress-induced glucocorticoids reduce BDNF expression and impair synaptic plasticity and memory in the hippocampus [38,39,40]. ASH has effects on the CNS, including a protective effect against neurotoxicity, resulting in increased BDNF mRNA levels [41,42,43,44,45]. Thus, anti-depressive effects of ASH may occur through the regulation of BDNF expression. However, changes in hippocampal BDNF signaling due to ASH have not been thoroughly investigated.

Therefore, in this study, we examined the anxiolytic effect of ASH using the novelty suppressed feeding (NSF) test and the improved elevated beam walking (IEBW) test, which is an improved version of the elevated plus maze (EPM), using electrocardiogram (ECG) analysis. Simultaneously, we conducted Western blot and immunohistochemistry analyses to explore potential mechanisms involving BDNF signaling that may underlie the anti-anxiety effects of ASH.

## 2. Results

### 2.1. NSF Test

#### 2.1.1. Behavior in the NSF Test

In this study, in rats housed under home cage (HC) housing conditions, we found no differences in the latency time to start eating 1% and 5% ASH extract-containing feed in the ASH1% and ASH5% groups compared with that in the Cont. group (Figure 1A). In contrast, under novel cage (NC) housing conditions, we observed a significant decrease in the latency time to start eating feed in the ASH1% and ASH5% groups compared with that in the Cont. group (*p* < 0.05; Figure 1A). In addition, when comparing HC and NC housing conditions, we found a significant increase in the latency time to start eating feed in both ASH1% and ASH5% groups under NC housing conditions (*p* < 0.05; Figure 1A). However, under HC housing conditions, there were no differences in food intake over 30 min in both groups (Figure 1B). Therefore, when we examined the frequency of pellet touches before the first continuous eating period exceeding 1 min under HC housing conditions, we found no change in frequency in the ASH1% and ASH5% groups (Figure 1C). In addition, when comparing HC and NC under HC housing conditions, the frequencies of the Cont. and ASH1% groups were significantly increased under NC housing conditions (*p* < 0.05; Figure 1C).

#### 2.1.2. ANS Activity (Heart Rate Variability)

Under HC housing conditions, we observed no differences in the LFnu (normalized unit of low frequency components), indicative of sympathetic nervous activity; HFnu (normalized unito of low frequency components), indicative of parasympathetic nervous activity; and LF/HF index of the cardiac sympathetic–parasympathetic tone balance values for all groups (Figure 2). In contrast, under NC housing conditions, the ASH5% group demonstrated significantly increased HFnu and significantly decreased LF/HF and LFnu values (*p* < 0.05; Figure 2A–C).

### 2.2. Comparison of Three Conditions in Heart Rate Variability

On the EPM test, the LF/HF and LFnu values tended to increase, but this autonomic change was unstable or statistically insignificant compared to HC housing conditions (Figure 3A–C). In contrast, LF/HF and LFnu values significantly increased compared to HC housing conditions on the IEBW (Figure 3A,B).

### 2.3. IEBW Test

#### 2.3.1. Time Spent in the Open Arm of the IEBW

In this study, the staying time in the open area of the IEBW significantly increased in the ASH1% and ASH5% groups compared to the Cont. group (*p* < 0.05; Figure 4).

#### 2.3.2. ANS Activity (Heart Rate Variability)

In this study, we placed rats implanted with the telemetry radio under HC housing conditions and measured the ANS activity at rest. As a result, in each conditioned rat, we observed no effects of the treatment by each reagent on the ANS and LF/HF values were low. Subsequently, we measured the ANS activity under IEBW conditions. As we observed under HC housing conditions, we found no effects of the treatment by each reagent on the ANS in each rat. However, under IEBW conditions, LFnu values were higher compared to HC housing conditions. Accordingly, we also observed an increase in LF/HF values (Figure 5A). Thereafter, the rats were orally administered the corresponding reagents for each treatment group and the ANS activity at rest measured. As a result, HFnu and LFnu values of the ASH1% and ASH5% groups significantly increased and decreased, respectively, compared to the Cont. group (Figure 5B). Furthermore, LF/HF values of the ASH1% and ASH5% groups significantly decreased compared to the Cont. group (Figure 5).

### 2.4. Western Blotting

In ASH1% or ASH5% groups, we dissected the hippocampus from the brain of each rat and detected BDNF signaling by Western blotting. The hippocampal BDNF protein expression levels in the ASH1% group were the same as those observed in the Cont. group, but significantly increased in the ASH5% group (Figure 6A). Moreover, in the ASH5% group, the phosphorylation of tropomyosin receptor kinase B (TrkB) as a BDNF receptor was significantly elevated, followed by a significant increase in the downstream phosphorylation of cAMP response element binding protein (CREB) of the TrkB cascade (Figure 6B,C).

### 2.5. Immunohistochemistry

Next, we examined potential immunohistological changes in the BDNF protein in the hippocampus of the ASH5% group. We observed a marked increase in BDNF-positive protein in the rat hippocampus in the case of ASH5% oral administration (Figure 7).

## 3. Discussion

Previous studies showed that ASH administration has a protective effect against gastric ulcers induced by cold-water restraint [12]. In addition, studies showed that ASH administration prevents depressive symptoms induced by tail suspension and the forced swimming test, with decreased immobility time [13,14,15]. Consequently, ASH is presumed to have an anti-stress effect that prevents gastric ulcers and depression. It is generally known that excessive stress causes or exacerbates gastric ulcers and depression. According to the general adaptation syndrome postulated by Hans Selye, stress affects the hypothalamus, and both sympathetic-adrenergic and pituitary-adrenal cortex systems induce gastric ulcers and depression [28]. Stress-induced diseases develop with stress stimulation to the CNS as the basic axis. In addition, ASH appears to prevent stress-induced disease states, so we can infer the possibility of ASH also affecting the CNS. However, the potential effects of ASH on the CNS have not been sufficiently studied.

On the other hand, stress-induced gastric ulcers develop because of stress and its associated anxiety [46]. This mental anxiety is known to increase SNS activity, while decreasing PNS activity [32]. Therefore, it is important to analyze the effects of ASH on the ANS. In this study, we investigated the effects of ASH on the ANS in rats under both novel environmental stress and high-fear stress conditions, as well as on the CNS in rats.

Rats feel anxiety in novel environments. Therefore, even if rats are fed, the confirmation work of the bait becomes longer and, as a result, the latency time to start eating becomes longer. As such, as the latency time is shortened by the chronic administration of antidepressants, the chronic administration of antidepressants is then assessed by the NSF test [47]. Therefore, in this study, the NSF test was conducted to evaluate the anti-anxiety and/or anti-stress effects of ASH. Before starting the NSF test, to normalize the NSF test results, ASH-administrated rats were placed under HC conditions. As a result, even when ASH was administered under HC housing conditions, no changes were observed in any of the consideration items examined, including the latency time to start eating, food intake, or the number of pellet touches before the first continuous eating of >1 min (Figure 1). In addition, ASH administration did not affect the ANS. We also found that ASH does not affect the ANS and causes no behavioral changes under normal conditions without stress from these events. Thus, by administering ASH to rats under stressed conditions, it was possible for us to examine the anti-anxiety and anti-stress effects of ASH. Subsequently, we examined the effects of ASH under NC conditions, and found that the latency time to start eating and the number of pellet touches before the first continuous eating period of >1 min significantly increased by the novel environmental change, but significantly decreased by ASH5% administration. In addition, the latency time to start eating significantly decreased, even with ASH1% administration (Figure 1A). It is known that the latency time to start eating in rats becomes significantly longer when the rats are in novel environments, which is consistent with our findings [47]. Concomitantly, we confirmed that the number of pellet touches before the first continuous eating period of >1 min significantly increased (Figure 1C). These behaviors are attributed to the increased alertness caused by anxiety. Thus, our results confirmed that this alertness is relieved by ASH administration, which may suppress anxiety-induced alertness (Figure 1). In addition, because ASH decreased SNS activity and increased PNS activity under NC housing conditions, the anxiolytic action of ASH is supported (Figure 2). Benzodiazepines have already been reported to reduce the latency time to start eating in novel environments [47].

Next, we examined the effects of ASH on novel environmental stress and high-fear stress. It is known that environmental factors, such as increased fear of heights, activate the SNS and inhibit the PNS, resulting in various anxiety-related behaviors and impaired physical condition [46,48,49]. In addition, patients suffering from anxiety disorders have excessive SNS activity and PNS inactivation, and similar symptoms are observed even in rodents [31,32,33]. Thus, rats undergoing high-fear stress can become animal models of anxiety disorders. Therefore, we analyzed the ANS of the rats by ECG under a high-fear stress environment. To this end, we first conducted an EPM test, which is generally used as an anxiety-related behavior assessment method. We found that the ANS score showed a tendency to increase, but not significantly, under HC housing conditions compared to the Cont. group (Figure 3). We then performed an IEBW test, in which we increased the height to induce a stronger fear stress. As a result, LFnu values, which are a parameter of SNS activation, significantly increased and the balance state of LF/HF became excited (Figure 3). This correlated with the autonomic balance observed in typical anxiety disorders [32]. Therefore, in this study, we used IEBW as an assessment method of anxiety. The staying time in the open arm of the IEBW, which strengthened the height fear stress, is a parameter used to evaluate the degree of anxiety. Then, when the Cont. and ASH groups were each placed in the IEBW, a significant extension of the staying time in the open arm was observed in the ASH group (Figure 4). Next, the effects on the ANS were examined. Under HC conditions, we confirmed that HFnu values, an indicator of PNS activity, remained stable at a high level in all the rats in each treatment group. In addition, the ANS was also stable from the LF/HF index (Figure 5). However, when rats were placed in the IEBW environment, LFnu values, an index parameter of SNS activity, increased and the rats became more excited. This was also confirmed from the LF/HF index. Thereafter, when ASH was administered orally and the same IEBW test was carried out, the rat SNS activity, which had been rising, significantly decreased by the administration of ASH1% or ASH5%. Accordingly, the rat PNS activity, which had decreased, increased with ASH1% or ASH5% administration (Figure 5). This process led to significant reductions (to a state close to rest) in the LF/HF index of rats that had been rising because of IEBW. These results suggest that ASH has an anxiolytic effect against not only mild anxiety, but also anxiety due to higher levels of stress, such as when the ANS is severely affected.

Recent studies reported that BDNF/TrkB signaling activation is important for ameliorating the depression-like pathology observed by the NSF test, and it can be improved by increasing BDNF expression in the hippocampus [50,51]. In addition, studies have reported that increased BDNF activity in the hippocampus has an anxiolytic effect [52]. Moreover, hippocampal BDNF mRNA levels reportedly decrease as a result of both acute and chronic stress [53]. These findings suggest that stress- and anxiety-related behaviors caused by stress are closely related to changes in hippocampal BDNF expression. In addition, CREB phosphorylation plays an important role in mediating BDNF responses in neurons [54]. Furthermore, a previous study reported the inducement of a neuroprotective effect through BDNF mRNA levels in PC12 cells [41]. It is possible that the anxiolytic effects due to ASH administration under high-level stress conditions may be due to changes in BDNF expression. Therefore, we conducted Western blot analyses of BDNF-/TrkB-related proteins in the hippocampus, along with an immunostaining analysis using a specific anti-BDNF antibody from ASH-administered rat brains from the IEBW test. Our results revealed that when ASH5% is administered to rats under IEBW conditions, the protein expression levels of BDNF in the rat hippocampus significantly increase and the phosphorylation of TrkB, and CREB also significantly increases. Furthermore, immunohistochemical analysis revealed that BDNF expression in the rat hippocampus is markedly elevated under IEBW conditions when treated with ASH5%. These behavioral and pharmacological experimental results obtained by ASH administration, including the resultant changes in BDNF expression, suggest that the effects of ASH on ANS activity occur via the hippocampus. This is strongly supported by the observation that the ventral part of the hippocampus is known to be involved in the defense reaction against anxiety [55].

Anxiety is also reported to be related to emotional eating in obese individuals [56,57]. In addition, hippocampal BDNF plays important role in food intake and body weight regulation [58]. Furthermore, the harmful effect of high-fat diet on learning and memory is linked to decrease in the hippocampal BDNF expression [58]. This study suggests that the anxiolytic effect and activation of the BDNF signaling by ASH extract improve the exacerbation in obesity-related pathological changes.

## 4. Materials and Methods

### 4.1. Plant Extract

A dried 2–5 cm root tip of ASH from Heilongjiang, China, was cut and extracted with hot water. The extract was concentrated under reduced pressure and dried using a spray dryer. The herb specimen was authenticated by Emeritus Professor Sansei Nishibe of the Laboratory of Pharmacognosy, Department of Pharmaceutical Sciences, Health Science University of Hokkaido. The extract powder (Lot No. 8142) was provided by Sun Chlorella Co., Ltd. (Kyoto, Japan). We measured each major component in the extract using HPLC and an ODS column. ASH extracts primarily comprised isofraxidine (101.4 mg/100 g), eleutheroside B (325.2 mg/100 g), eleutheroside E (625.2 mg/100 g), eleutheroside B1 (95.2 mg/100 g), and chlorogenic acid (829.5 mg/100 g).

### 4.2. Ethics Statement

This study was conducted according to the “Guide for the Care and Use of Laboratory Animals” (NIH Publication No. 85–23, revised in 1996). All experimental protocols were approved by the Ethics Committee on Animal Use of the Suzuka University of Medical Science (No. 1 of April 1, 2016).

### 4.3. Animals

Male Sprague-Dawley (SD) rats (6 weeks old) were purchased from SLC, Inc. (Shizuoka, Japan), individually housed in standard polycarbonate cages for 7 days, and subjected to serial 7-day handling. A normal diet (ND: MF, Oriental Yeast Manufacturing Co., Ltd., Tokyo, Japan) and water were available *ad libitum*. Rats were kept in a room maintained at 23 ± 2 °C and 50–65% humidity under a 12 h light/dark cycle (lights on at 7:00).

### 4.4. Surgery

After a 1-week acclimatization, rats were treated with a local anesthetic (ropivacaine hydrochloride hydrate) and general anesthesia (sodium pentobarbital, 40 mg/kg, i.p.), and a wireless telemeter (model TR50BB; Biotelemetry Research, Phoenix, AZ, USA) was implanted in the abdomen. Animals were then intramuscularly injected with antibiotics (imipenem hydrate and cilastatin sodium, 8.3 mg/kg) to prevent postoperative infections. After the operation, rats were allowed to recover for 7 days.

### 4.5. Administration

Rats were divided into three groups [control (Cont.), ASH1%, and ASH5%] based on their body weight before administration. Test foods were prepared by mixing ND with the ASH extract and were provided ad libitum for 5–7 days. In addition, 1 mL of distilled water, or ASH1% and ASH5% dissolved in distilled water, was administered through a probe only once 30 min before each behavioral test.

### 4.6. Behavioral Studies

#### 4.6.1. NSF Test

The NSF test is often used as a measure of depression/anxiety-like behaviors and was conducted as described in Bodnoff et al. [47], with slight modifications. The NSF test schedule is described in Figure 8A. For the NC, we used a cage where the bottom and sides were covered with a black plastic garbage bag without bedding (NC; Figure 8B). Food-deprived (24 h) animals were subjected to a 30-min HC (Figure 8B, HC: 27 × 43 × 20 cm^3^) test after placing pellets on the wire netting of the cage in the housing room and in the NC test by the same method after a 1-day interval. Pellets were given to rats during the interval period. We assessed latency time to start eating, the number of pellet touches before the first continuous eating period of >1 min, and food intake. HRV analysis was conducted in Cont. and ASH5% groups.

#### 4.6.2. IEBW Test

In order to examine the anxiolytic effect of ASH, we carried out behavioral experiments using high-fear stress. In generally, the EPM test was conducted to evaluate animal anxiety [59]. We observed stable ANS activity in the HC (Figure 9A). Only those rats in which the ANS was not disturbed by the experimenter under normal conditions were included. However, in the EPM apparatus 50 cm above the floor (Figure 9B) for 3 min, the LF/HF and LFnu values tended to increase, but this ANS change was unstable or statistically insignificant compared to HC housing conditions (Figure 3A–C). In contrast, LF/HF and LFnu values significantly increased compared to HC housing conditions on the IEBW (Figure 3A,B). Thus, we used the IEBW apparatus (Figure 9C) by installing a 2 × 4 timber (180 × 8.9 cm^2^) 190 cm above the floor level and an open (140 × 8.9 cm^2^) and closed (40 × 8.9 × 28.5 cm^3^) arms, which improved the EPM apparatus. The IEBW test schedule is described in Figure 9D. After a 7-day administration, they were placed in the tip of the open arm 140 cm away from the closed arm, and their behavior and HRV were examined for 3 min from 0.5 h after oral administration (IEBW test). We then assessed the time spent in the open arm, LF/HF, LFnu, and HFnu.

### 4.7. Brain Tissue Preparation

After behavioral tests, rats were euthanized, and their brains were rapidly excised. Hippocampal tissues were stored at −70 °C until Western blotting.

### 4.8. Assessment of Cardiac Autonomic Activity (Heart Rate Variability)

After the operation, ECG data were continuously recorded using the radio-telemetry system, comprising the implanted device, a battery-charging and telemetry-receiving device underneath the cage, and a data acquisition system (PowerLab16/35, PL3516, AD Instruments, Castle Hill, New South Wales, Australia) interfaced with a computer. Cardiac autonomic activity was assessed by spectral analysis of R-R interval variability. Data were stored and analyzed using LabChart Pro (ver. 8.0, AD Instruments). Frequency domain analysis and power spectra of R-R interval variability were obtained using the fast Fourier transform algorithm. High (HF: 0.6–3.0 Hz), low (LF: 0.2–0.6 Hz), and very low (VLF: ≤0.2 Hz) frequencies were determined. LF and HF components were expressed in normalized units (LFnu and HFnu). The power of the HF component indicates cardiac vagal activity, whereas the LF component indicates the sympathetic activity with vagal modulation. LF/HF is an index of the cardiac sympathetic-parasympathetic tone balance.

### 4.9. Western Blotting

Homogenized hippocampal lysates were separated by 10% sodium dodecyl sulfate-polyacrylamide gel electrophoresis and transferred to Immun-Blot PVDF membranes (Bio-Rad, Hercules, CA, USA). Membranes were incubated with 3% bovine serum albumin (BSA) in TBST for 1 h, followed by incubation with primary antibodies against BDNF (ab108319), TrkB; #4603S), p-TrkB (#4619S), p-CREB (#9198S), CREB (#9197S), and β-actin (#4970S) at a 1:1000 dilution overnight at 4 °C. Anti-BDNF antibody was bought from Abcam (Cambridge, UK), and other primary antibodies were bought from Cell Signaling Technology Japan, K.K. (Tokyo, Japan). This was followed by incubation with horseradish peroxidase-conjugated secondary antibodies (#7074S, 1:1000, Cell Signaling Technology Japan) for 1 h. Immunoreactive bands were detected using an enhanced chemiluminescence detection kit, and a Light Capture AE-6971/2 device (ATTO Corp., Tokyo, Japan) was used for visualization. Band intensities were normalized to β-actin, total TrkB, or CREB using CS Analyzer 4 (ATTO Corp.).

### 4.10. Immunohistochemistry

Rat brains were fixed overnight by 4% paraformaldehyde, followed by immersion in graded sucrose solutions for cryopreservation (10%, 20%, and 30% sucrose in phosphate-buffered saline [PBS]) and freezing with dry ice. Frozen brains were mounted using Tissue-Tek^Ⓡ^ O.C.T. compound (Sakura Finetek Japan Co., Ltd., Tokyo, Japan) and stored at −80 °C. Brains were then sectioned into 10-µm thick sections using a cryostat at −15 °C. Sections were collected in PBS and incubated with 10% BSA in PBS for 1 h, followed by incubation with anti-BDNF antibody (sc-20981, 1:100, Santa Cruz Biotechnology, Santa Cruz, CA, USA) for 3 days at 4 °C. Incubation with the secondary antibody (#4413S, 1:1000, Cell Signaling Technology Japan) proceeded for 2 h, after which sections were mounted onto MAS-coated glass slides and cover-slipped using Fluoromount (Diagnostic BioSystems, Pleasanton, CA, USA). Images were obtained with a fluorescence microscope (BZ-9000, Keyence, Osaka, Japan).

### 4.11. Statistics Analysis

Values are expressed as means ± standard errors and were derived from measurements of 4–8 rats (NSF test, *n* = 6; IEBW test, *n* = 8, Western Blotting, *n* = 4–5). Statistical analysis was performed using SPSS statistics 24 (IBM, Tokyo, Japan). Homogeneity of variances was checked with the Levene’s test. One-way ANOVA was used for inter-group comparisons. When ANOVA revealed significant differences, Dunnett’s *t* or T3 *post hoc* tests were used to identify significant differences from the Cont. group. Differences between conditions were analyzed using repeated measures ANOVA. Differences between two groups were analyzed by paired Student’s *t*-test. *P-*values < 0.05 were considered significant.

## 5. Conclusions

Our novel findings revealed a unique regulation of the balance of the ANS based on parasympathetic dominance and the anxiolytic actions of ASH. In addition, our results suggested that an increase in hippocampal BDNF signaling is involved in the actions of ASH. Collectively, our findings demonstrated that ASH extract with the anxiolytic effects might be a beneficial supplement or preventive medicine for the maintenance of mental health or against obesity associated with emotional eating.

## Figures and Tables

**Figure 1 molecules-24-00132-f001:**
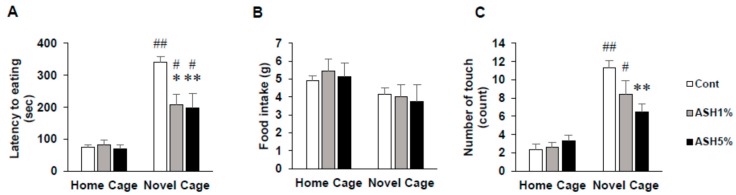
Effects of *Acanthopanax senticosus* HARM (ASH) on anxiety-related behaviors induced by mild stress in the novelty suppressed feeding (NSF) test. (**A**) Latency time to start eating. (**B**) Total food intake. (**C**) The number of pellet touches before the first continuous eating period of >1 min. Data are presented as mean ± SE; *n* = 6; statistical significances are denoted by * *p* < 0.05 and ** *p* < 0.01 compared with the novel cage (NC) Cont. group (Dunnett’s *t*-test), ^#^
*p* < 0.05 and ^##^
*p* < 0.01 compared with the home cage (HC) group (paired *t*-test).

**Figure 2 molecules-24-00132-f002:**
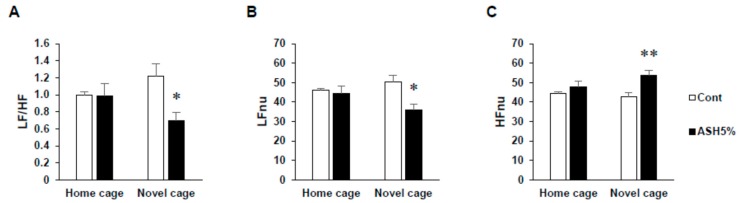
Effects of ASH on the autonomic nervous control of heart rate in the novelty suppressed feeding (NSF) test. Power spectrum analysis of heart rate variability in the NSF test. (**A**) LF/HF indicates the balance of the sympathetic and parasympathetic nervous systems. (**B**) LFnu (normalized unit of the LF value) indicates the sympathetic nervous system activity. (**C**) HFnu (normalized unit of the HF value) indicates the parasympathetic nervous system activity. Data are presented as mean ± SE; *n* = 6; statistical significances are denoted by * *p* < 0.05 and ** *p* < 0.01 compared with the NC Cont. group (unpaired *t*-test).

**Figure 3 molecules-24-00132-f003:**
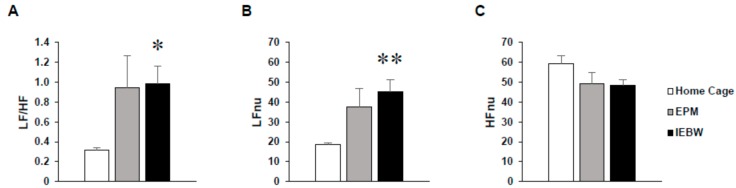
Analysis of the HRV in the HC and on the EPM and IEBW. (**A**) LF/HF indicates the balance of the sympathetic and parasympathetic nervous systems. (**B**) LFnu (normalized unit of the LF value) indicates the sympathetic nervous system activity. (**C**) HFnu (normalized unit of the HF value) indicates the parasympathetic nervous system activity. Data are presented as mean ± SE; *n* = 6; statistical significances are denoted by * *p* < 0.05 and ** *p* < 0.01 compared with the HC group [repeated measures analysis of variance (ANOVA)].

**Figure 4 molecules-24-00132-f004:**
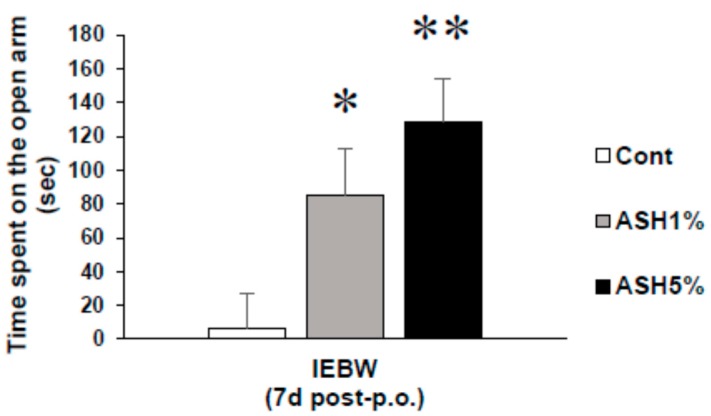
Effects of ASH on anxious behaviors induced by strong stress in the IEBW test. Time spent on the open arm. Data are presented as mean ± SE; *n* = 8; statistical significances are denoted by * *p* < 0.05 and ** *p* < 0.01 compared with the Cont. group (Dunnett’s T3 test).

**Figure 5 molecules-24-00132-f005:**
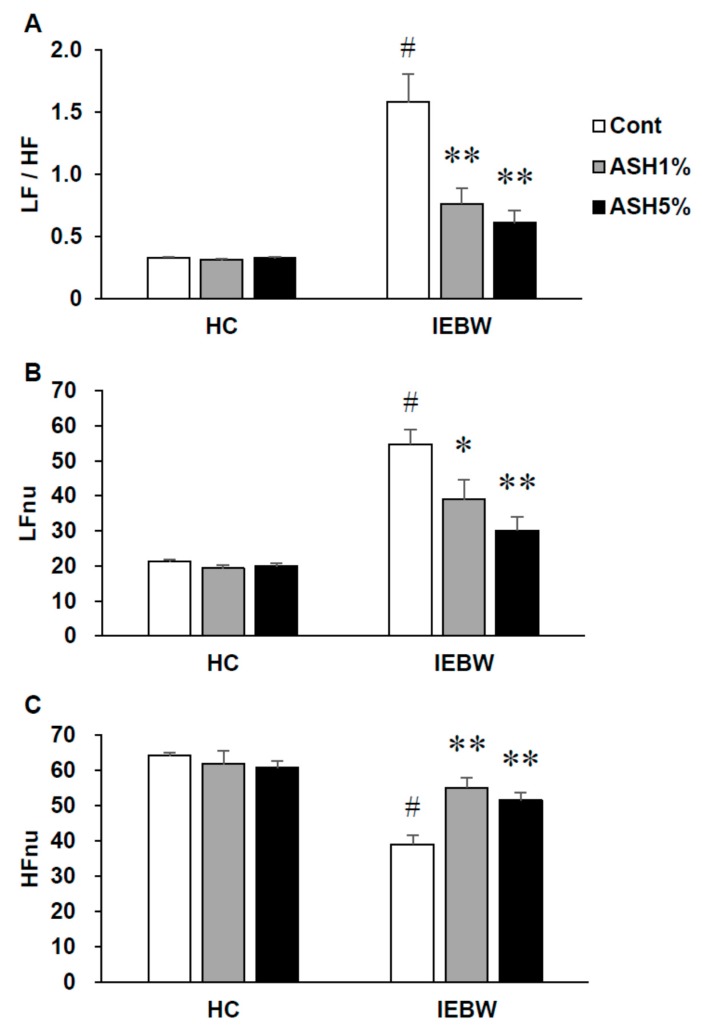
Effects of ASH on the autonomic nervous system control of heart rate in the IEBW test. The power spectrum analysis of heart rate variability (HRV) on IEBW. (**A**) LF/HF indicates the balance of the sympathetic and parasympathetic nervous systems. (**B**) LFnu (normalized unit of the LF value) indicates the sympathetic nervous system activity. (**C**) HFnu (normalized unit of the HF value) indicates the parasympathetic nervous system activity. Data are presented as the mean ± SE; *n* = 8; statistical significances are denoted by * *p* < 0.05 and ** *p* < 0.01 compared with IEBW Cont. (Dunnett’s *t*-test (^#^
*p* < 0.05 compared with the HC group (paired *t*-test).

**Figure 6 molecules-24-00132-f006:**
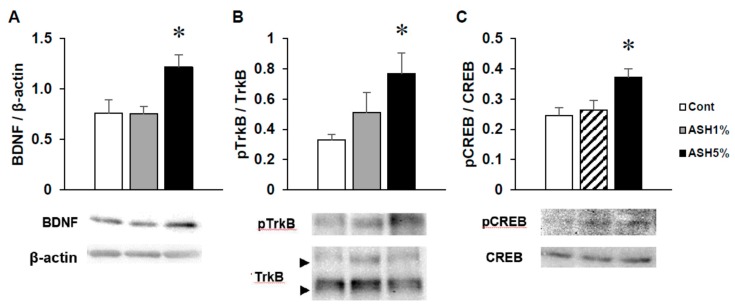
**Effects of ASH on hippocampal BDNF signaling.** Analysis of hippocampal (**A**) BDNF, (**B**) phosphor-TrkB (p-TrkB), and (**C**) p-CREB protein levels by Western blotting. BDNF was normalized to β-actin, whereas p-TrkB and p-CREB were normalized to total TrkB and CREB. Each data item showed the ratio vs. Cont. Data are presented as mean ± SE; *n* = 4–5; statistical significances are denoted by * *p* < 0.05 compared with the Cont. group (Dunnett’s *t*-test).

**Figure 7 molecules-24-00132-f007:**
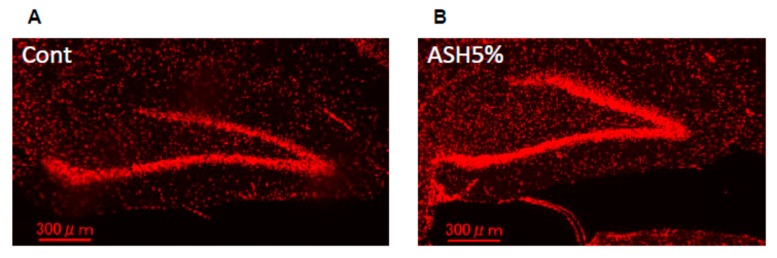
Immunohistochemical staining for hippocampal BDNF. Representative images of staining with the anti-BDNF antibody in (**A**) Cont. and (**B**) ASH5%.

**Figure 8 molecules-24-00132-f008:**
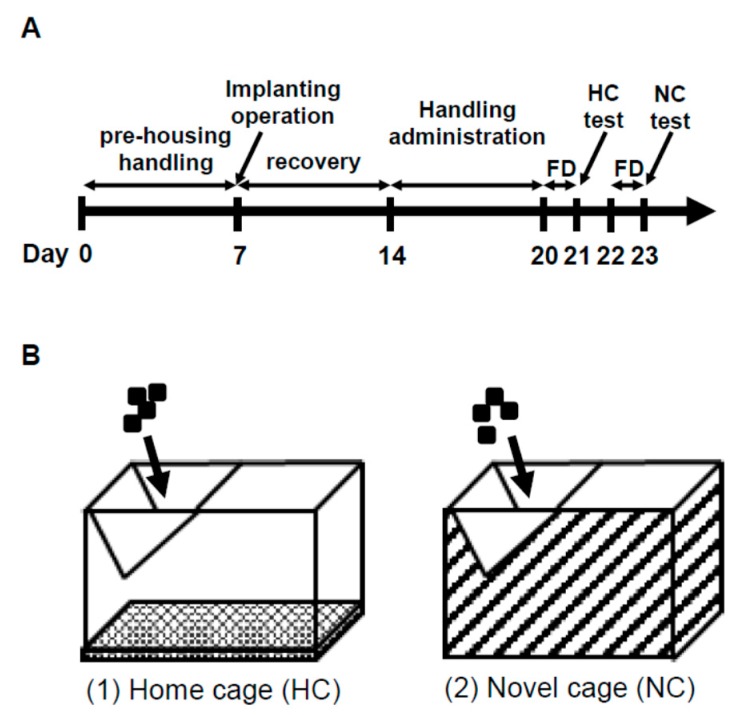
Schedule and test apparatus in NSF test. (**A**) Schedule of the NSF test. FD: Food deprivation. (**B**) Modified model of the NSF apparatus. (1) Home cage. (2) Novel cage.

**Figure 9 molecules-24-00132-f009:**
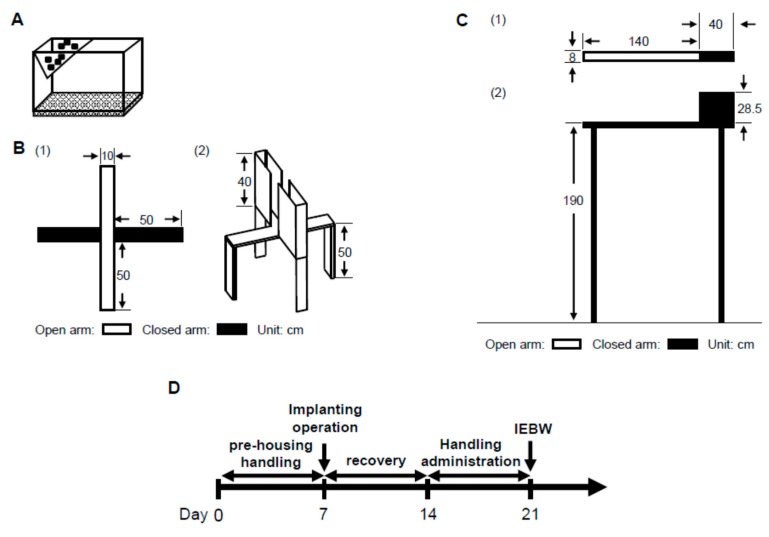
Improved elevated beam walking (IEBW) as an improved version of the elevated plus maze (EPM). The three apparatuses: (**A**) Home cage (HC); (**B**) EPM, (1) view from above and (2) bird’s eye view; and (**C**) IEBW (open arm has no wall and closed arm has wall), (1) view from above and (2) view from the side. (**D**) IEWB test schedule.

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
