# Peer review of "Anxiolytic Effects of Acanthopanax senticosus HARMS Occur via Regulation of Autonomic Function and Activate Hippocampal BDNF–TrkB Signaling"

_molecules, 2018, doi:10.3390/molecules24010132_

Reviewer 1 Report

The present manuscript aimed to investigate the potential effect of Acanthopanax senticosus HARMS (ASH) on mitigating the anxiety caused by environmental stress in rodent model and to reveal the mechanisms underlying the anxiolytic effect. In addition to identify the potentially useful agents for maintaining good mental health, this study is also the first report to demonstrate that the anxiolytic effects of ASH were resulted from the regulation of autonomic function and increased hippocampal BDNF signaling. This submitted manuscript may fit the quality and scope of the journal and accept for publication, however, it needs major modifications as follows.

1. The text should be written in scientific English. It is strongly recommended to revise the manuscript by native English speakers before resubmitting.

2. Figure 1A: How to rule out the reduced latency to eating is not attributed to the smell of ASH in the novel cage?

3. Figure 2: The statistical symbol # should be omitted from the legend since there is no such symbol in Figure 2A to 2C.

4. Section 2.2.2 Time spent in the open arm of the IEBW and Figure 4: Is it possible that the rats exhibit fear behavior so that the latency to reach the closed arm increased due to they are immobile?

5. Figure 6: Generally, proteins are illustrated as black bands for Western Blotting, thus the images should be inverted.

6. Figure 6A: The intensity for β-actin is stronger than that of BDNF; therefore, the ratio of BDNF/β-actin should not be over 1. The authors need to replace the image of western blot or re-do the densitometric analysis.

7. Figure 6B: There are two TrkB bands in Figure 6B. Which one is used to calculate the ratio of p-TrkB/TrkB?

8. Section 4.5 Administration: The description of ASH administration is confusing as ASH was mixed with normal diet and ASH was dissolved in distilled water as well. The actual route for ASH administration should be addressed clearly.

9. Section 4.7.1 NSF test: The figure numbers in the context should be labeled as Figure 8A and B(2).

10. Section 4.7.4 NSF test: The authors did not switch the order of HC and NC during the test. How to make sure that it won’t be interfere with the results since rats may learn the feeding experience?

11. Section 4.7.2 IEBW test: The figure numbers in the context should be labeled as Figure 9A, 9B, 9C and 9D.

12. Section 4.7.2: Is there any evidence to prove IEBW is more suitable for testing anxiety than EPM and/or beam walking?

Author Response

We thank referees for careful reading our manuscript and for giving useful comments.

Our responses to the referee's comments are as follows:

1. The text should be written in scientific English. It is strongly recommended to revise the manuscript by native English speakers before resubmitting.

The manuscript was proofreading by native speakers of English.

2. Figure 1A: How to rule out the reduced latency to eating is not attributed to the smell of ASH in the novel cage?

We could not eliminate the influence of smell. Volatile components may affect the latency time to eating in the novel cage.

3. Figure 2: The statistical symbol # should be omitted from the legend since there is no such symbol in Figure 2A to 2C.

Symbol # was omitted from the legend of Figure 2A-C.

4. Section 2.2.2 Time spent in the open arm of the IEBW and Figure 4: Is it possible that the rats exhibit fear behavior so that the latency to reach the closed arm increased due to they are immobile?

We observed that rats moved on the open arm (i.e., turning around, sniffing, and walking). Thus, we considered the increased time spent on the open arm to be due to the anxiolytic potential of ASH.

5. Figure 6: Generally, proteins are illustrated as black bands for Western Blotting, thus the images should be inverted.

The images of bands were accordingly inverted.

6. Figure 6A: The intensity for β-actin is stronger than that of BDNF; therefore, the ratio of BDNF/β-actin should not be over 1. The authors need to replace the image of western blot or re-do the densitometric analysis.

We reanalyzed densitometry and replaced the image as suggested.

7. Figure 6B: There are two TrkB bands in Figure 6B. Which one is used to calculate the ratio of p-TrkB/TrkB?

According to the data sheet, both bands are derived from TrkB. Thus, we calculated the sum of the two bands as TrkB.

8. Section 4.5 Administration: The description of ASH administration is confusing as ASH was mixed with normal diet and ASH was dissolved in distilled water as well. The actual route for ASH administration should be addressed clearly.

Thank you for the comment. We have revised this description. Please check whether it is appropriate.

9. Section 4.7.1 NSF test: The figure numbers in the context should be labeled as Figure 8A and B(2).

We relabeled the number.

10. Section 4.7.4 NSF test: The authors did not switch the order of HC and NC during the test. How to make sure that it won’t be interfere with the results since rats may learn the feeding experience?

There were three reasons why we did not switch the order of HC and NC during the test as follows:

We considered that analysis of the hippocampus of the rats under stress is important for obtaining correct information; therefore, we conducted the sampling immediately after the NC test.

We attempted to reduce the animal number as much as possible in accordance with the policy of 3Rs; thus, we did not conduct the additional experiment for sampling alone.

We could not deny the interference by feeding experience, although Silke et al. have reported no significant difference between the fixed-time feeding and food deprivation groups in the latency to eat (Silke D.; Katarina R. L.; Heidrun F.; Jan B.; Jörg-Peter V. Food Deprivation, Body Weight Loss and Anxiety-Related Behavior in Rats. Animals. 2016, 6(1), 4, DOI:10.3390/ani6010004).     

11. Section 4.7.2 IEBW test: The figure numbers in the context should be labeled as Figure 9A, 9B, 9C and 9D.

We have relabeled the numbers.

12. Section 4.7.2: Is there any evidence to prove IEBW is more suitable for testing anxiety than EPM and/or beam walking?

In this study, we analyzed the HRV of rats under three conditions. This result (Section 2.2.) shows that the error of autonomic nervous system activity value is smaller on the IEBW test than on the EPM test.

Reviewer 2 Report

Lines 40-44: The description of plant, and their usaes, applications and properties should be improved. In addition the description of profile of bioactive compounds should be inserted. The aim of study should be rewritten.

In Material and Methods major details of plant extract should be given. The Material and Methods should be organized in two main sections: Design of study and analytical procedures.

The statistical analysis is not clear as reported in histograms.In Results, the structure of paragraph and subparagraph is not clear.

In the Discussion the authors should better compare the results with previous data in literature data.

More recent references along the whole manuscript should be inserted.

Conclusion should be implemented.

Author Response

We thank referees for careful reading our manuscript and for giving useful comments.

Our responses to the referee's comments are as follows:

Lines 40-44:

The description of plant, and their usages, applications and properties should be improved.

We have added the description about their usages and applications (Lines 4145).

In addition, the description of profile of bioactive compounds should be inserted.

We have included the description of the profile of bioactive compounds (Lines 4956).

The aim of study should be rewritten.

We have rewritten the aim of the study (Lines 7882).

In Material and Methods major details of plant extract should be given.

In the description of Section 4.1, we have included the extraction method and result of measuring major components by HPLC. We would appreciate if you could specify what further information needs to be added here, if any.

The Material and Methods should be organized in two main sections: Design of study and analytical procedures.

We have reconstructed the sections as follows:

Design of study: Sections 4.14.7., analytical procedures: Sections 4.84.11.

The statistical analysis is not clear as reported in histograms. In Results, the structure of paragraph and subparagraph is not clear.

We have revised the description of the legend and reconstructed the Results section.

In the Discussion the authors should better compare the results with previous data in literature data.

Jin, L. et al. have reported no significant difference in the number of crossings and rearing in the open-field behavior test compared with that in the control group, although their study did not show the data ( [14]). Thus, we did not compare these results.

Wu, F. et al. reported ASH induced BDNF mRNA in the PC12 cells ( [41]). Our study showed ASH induced hippocampal BDNF protein expression for the first time. Thus, we could not compare the results with previous data in the literature.

More recent references along the whole manuscript should be inserted.

We have inserted some additional recent references.

Conclusion should be implemented.

Please check Section 5 for conclusions.

Round  2

Reviewer 1 Report

The revised manuscript has been much improved and the comments have been responded solidly. Accordingly, it can be accepted for publication. I strongly recommend the authors could consider the following suggestions to enhance the quality of this manuscript.

1. It would be better to make a statement regarding the influence of ASH smell in the discussion.

2. The explanation of IEBW is more suitable for anxiety test is acceptable. The explanation could be added in section of 4.6.2 or in discussion to re-emphasize the advantage of IEBW.

Author Response

Response to Reviewer 1 Comments

The revised manuscript has been much improved and the comments have been responded solidly. Accordingly, it can be accepted for publication. I strongly recommend the authors could consider the following suggestions to enhance the quality of this manuscript. 

The author would like to thank the referee for useful suggestions that helped us to improve the original manuscript.

1. It would be better to make a statement regarding the influence of ASH smell in the discussion.

The first comment is a very sharp suggestion, and very useful remark suggesting the possibility of ASH's smell. However, in the literature search, there is no report on the smell of Acanthopanax senticosus HARM and quotation is impossible. 

2. The explanation of IEBW is more suitable for anxiety test is acceptable. The explanation could be added in section of 4.6.2 or in discussion to re-emphasize the advantage of IEBW.

The second comment by reviewer 1 is a very important suggestion to appeal the usefulness of IEBW developed by us. Therefore we modified the first half of section 4.6.2 as follows.

In order to examine the anxiolytic effect of ASH, we carried out behavioral experiments using high fear stress. In generally, the EPM test was conducted to evaluate animal anxiety [59]. We observed stable ANS activity in the HC (Figure 9A). Only those rats in which the ANS was not disturbed by the experimenter under normal conditions were included. However, in the EPM apparatus 50 cm above the floor (Figure 9B) for 3 min, the LF/HF and LFnu values tended to increase, but this ANS change was unstable or statistically insignificant compared to HC housing conditions (Figure 3A–C).In contrast, LF/HF and LFnu values significantly increased compared to HC housing conditions on the IEBW (Figure 3A and B).Thus, we usedthe IEBW apparatus (Figure 9C) by installing a 2 × 4 timber (180 × 8.9 cm2) 190 cm above the floor level and an open (140 × 8.9 cm2) and closed (40 × 8.9 × 28.5 cm3) arms, which improved the EPM apparatus.”

Reviewer 2 Report

The authors have improved the manuscript that it is now suitable for publication

Author Response

Response to Reviewer 2 Comments

The authors have improved the manuscript that it is now suitable for publication.

We would like to take this opportunity to thank you for taking the trouble of reviewing our manuscript. Thank you for considering our paper carefully and for valuable comments.